# Dynamical Systems Analysis of $f(Q)$ Gravity

Christian Böhmer [1] , Erik Jensko [1] and Ruth Lazkoz [2,*]

1   Department of Mathematics, University College London, Gower Street, London WC1E 6BT, UK
2   Department of Physics and EHU Quantum Center, Faculty of Science and Technology,
    University of the Basque Country, 48080 Bilbao, Spain
*   Correspondence: ruth.lazkoz@ehu.es

**Abstract:** Modified gravity theories can be used for the description of homogeneous and isotropic cosmological models through the corresponding field equations. These can be cast into systems of autonomous differential equations because of their sole dependence on a well-chosen time variable, be it the cosmological time, or an alternative. For that reason, a dynamical systems approach offers a reliable route to study those equations. Through a model-independent set of variables, we are able to study all $f(Q)$ modified gravity models. The drawback of the procedure is a more complicated constraint equation. However, it allows the dynamical system to be formulated in fewer dimensions than using other approaches. We focus on a recent model of interest, the power-exponential model, and generalize the fluid content of the model.

**Keywords:** modified gravity; non-metricity; dynamical systems

## 1. Introduction

We have an unprecedented understanding of the gravitational interaction as the main actor in the large-scale dynamics of the Universe, being responsible for the formation and evolution of structures on the largest scales. Einstein's theory of General Relativity (GR) accounts successfully for a vast array of gravitational phenomena [1–3]. Unfortunately, the so-called dark sector, i.e., dark matter and dark energy, represents a true challenge to an otherwise successful paradigm. The aim of solving these problems has motivated the consideration of slight modifications of GR compatible with observations. The geometry of the Universe is assumed to be the spatially flat Friedmann–Lemaître—Robertson–Walker (FLRW) line element, in agreement with most current observational data [2,4,5]. The FLRW model appears to be the best cosmological model available at the moment [6–16]. It takes the form $ds^2 = -N(t)^2 dt^2 + a(t)^2 \delta_{ij} dx^i dx^j$ with scale factor $a(t)$ and arbitrary lapse function $N(t)$. In what follows we can set $N(t) = 1$ without loss of generality, however, we note that some modified theories of gravity might not be compatible with this choice. Modifications of GR can help address observational tensions on the expansion rate of the universe as given by $H_0$, and the $S_8$ parameter characterizing linear matter fluctuations on the scale of $8\,\text{h}^{-1}$ [17–19].

Modified theories of gravity have been studied for almost as long as GR itself [20,21]. Possible modifications may bring extra geometrical structures, increase the number of dimensions, or introduce non-linearities into the Einstein–Hilbert action which is linear in curvature. Other options include non-minimal matter curvature couplings. The vast majority of modified gravity models being considered to fall somewhere in the above description [22–37].

Our work will deal with field equations that contain derivatives no higher than second order with respect to the independent variables. Due to the specific structure of the models, one can introduce the Hubble function so that all field equations take the form $E_i(H, \dot{H}, \Psi) = 0$ where we use $\Psi$ to symbolize the presence of any matter fields. The rather simple description covers many (if not all) second-order modified gravity theories, provided non-minimal couplings are excluded. Let us also mention that we assume the usual matter conservation equations hold for each fluid individually. This is an assumption on the

modified gravity theory in question and is linked to diffeomorphism invariance, and most models satisfy this assumption.

It is useful to cast the cosmological field equations into the form of a dynamical system, for a comprehensive summary of past work in the field see [38] and references therein. We briefly mention that the choice of variables can be problematic, something that is well-known in $f(R)$ gravity, see for instance [39–42], see also [43] for $f(T)$ gravity. Similar to these approaches we also tend to find complicated constraint equations when the most useful variables are chosen. However, it is quite remarkable that we can make a significant number of general statements about the system for arbitrary models. For example, all possible de Sitter points can be found by the introduction of convenient functions [44].

In the following, we deal with the class of modified gravity theories referred to generally as symmetric teleparallel gravity, or $f(Q)$ gravity [45–47], which have similarities with the $f(\mathbf{G})$ and $f(T)$ theories, see [35,36,45,48–50] for details. In fact, the geometric scalars of these theories all coincide in cosmology $Q = T = -\mathbf{G} = 6H^2$. The cosmological field equations of these theories read

$$6f'H^2 - \frac{1}{2}f = \rho\,, \tag{1}$$

$$(12H^2f'' + f')\dot{H} = -\frac{1}{2}(\rho + p)\,, \tag{2}$$

and can be written as a dynamic system using standard techniques. Our formulation allows us a simple comparison with the successful ΛCDM model. Its early-time behavior is dominated by radiation which is a saddle (or repeller) from the dynamical systems viewpoint, whereas the late-time asymptotic regime represents a (de Sitter) cosmological constant-dominated attractor. The matter dominated epoch is a transient situation (saddle point). The specific $f(Q)$ model we will study here displays properties compatible with the ΛCDM model.

## 2. Brief Review on the Standard Approach to the Construction of a Dynamical System

It is rarely easy to find appropriate variables to formulate modified gravity models as dynamical systems, again see [38]. Motivations to introduce different variables can vary, leading to distinctly different features [39–41]. Here we briefly run through the standard approach and show how it is equivalent to our formulation in one fewer dimensions.

A typical approach takes the Friedmann Equation (1) and divides by $6f'H^2$ (where the assumption $f' \neq 0$ must be made). This excludes trivial constant functions, but also the case where $f'$ passes through zero dynamically, which may be important. The typical variables are

$$X_i = \frac{\rho_i}{6f'H^2}\,, \qquad Y = \frac{f}{12f'H^2}\,, \tag{3}$$

where we are considering various matter-energy sources that may exist. As a reminder, General Relativity of course corresponds to $f(Q) = Q + 2\lambda_0$, where $\lambda_0$ stands for a cosmological constant term (with appropriate dimensions). With these variables, the Friedmann equation becomes the following constraint

$$\sum_i X_i + Y = 1\,. \tag{4}$$

The naturalness of the Friedmann constraint with this choice of variables is immediately apparent.

The next step is to introduce an additional variable to remove the explicit dependence on the Hubble function which would appear in the dynamical equations. A convenient choice for this extra variable is

$$Z = \frac{Q/Q_0}{1 + Q/Q_0} = \frac{H^2/H_0^2}{1 + H^2/H_0^2}\,, \tag{5}$$

and clearly $Z$ is positive and smaller than 1 for all times (that is $Q = 6H^2$ is non-negative and finite).

The constraint equation allows us to remove one of the independent variables so that we are left with as many independent variables as one plus the number of matter sources. However, this is not the end of the story: since $f$ is a function of $Q$ it can always be rewritten in terms of $Z$ for some given function. Therefore, $Y$ can be removed altogether. This is exactly the approach we will take in the following section. One can think of $Y$ as encapsulating information about the free parameters that can be found in $f$ (such as the important case of a cosmological constant) but which are not depicted by Z. Further details on the meaning of this will be discussed with a specific example.

In the usual fashion, one then uses the time variable $N = \log a$. We will not produce the explicit form of the dynamical equations $dX_i/dN$ and $dY/dN$, which in the of for a single matter source can be found in our work [35]. It can be seen that defining the function

$$m = 12H^2 (\log f')',$$

(6)

allows us to make very general statements about the stability of the system using Jacobian $N$ equations.

We mention again the possibility of taking advantage of the dependence between variables $Y$ and $Z$ to remove one extra dimension from the system, see also [43]. To this end, we use the fact that $Y = f/(2f'Q)$, which by virtue of the chain rule along with Equation (5) and the constraint $\sum_i X_i + Y = 1$ allows us to write

$$\sum_i \dot{X}_i = -\frac{\partial Y}{\partial Q} \frac{(Q_0 + Q)^2}{Q_0} \dot{Z}.$$

(7)

We must remember throughout that $Q$ can always be recast as a function of $Z$. The next step is to use (7) to replace $\dot{X}_i$ in the equation where it appears. Next, the constraint $\sum_i X_i + Y = 1$ must be used to remove $X_i$ and then it is necessary to remember that $Y$ is a function of $Z$. This gives us a system with one fewer dimension and slightly different equations for the evolution of the dynamical variables. It is not difficult to see the agreement between the two approaches (with different dimensionalities), but again, we suggest the reader to check [44].

It may at first appear that this equivalent formulation poses no benefits, as a model needs to be specified in order to extract any useful information from the system. This, however, is incorrect, and we shall see that even in the completely general case we can analyze the dynamical system to some degree. Moreover, phase space and stability analysis become much simpler in fewer dimensions, despite representing the same physics.

### 3. Dynamical Systems Formulation

*3.1. General Setup with Two Fluids*

Generalizing the formulation of the problem with a reduced dimensionality we can now tackle a two-fluid case in just two dimensions. Their energy densities will be $\rho_1$ and $\rho_2$ with the equation of state parameters $w_1$ and $w_2$. For this discussion we leave these equations of state parameters arbitrary, though we will later set $w_1 = 0$ and $w = 1/3$. These choices render the cosmological equations as

$$-\frac{f}{6H^2} - \frac{\rho_1}{3H^2} - \frac{\rho_2}{3H^2} + 2f' = 0,$$

(8)

$$(w_1 + 1)\rho_1 + (w_2 + 1)\rho_2 + 2\dot{H}(f' + 12H^2 f'') = 0.$$

(9)

We now define the following dynamical variables

$$X_i = \frac{\rho_i}{3H^2}, \text{ for } i = 1, 2,$$

(10)

$$Z = \frac{H^2/H_0^2}{1 + H^2/H_0^2}.$$

(11)

The first two variables are non-negative, have an easily recognizable form (being simply the standard matter density parameters $\Omega_i$) and are different from their previous counterparts (3) as $f'$ does not appear in the denominator. There is now no dependence on $f$ in any of the variables. The reason for this is the ability to write any function of $H$ as a function of $Z$, with its dynamics being determined by the Friedmann constraint itself.

It is now possible to cast the Friedmann constraint as an expression that involves only the new variables

$$X_1 + X_2 = \left(1 - \frac{1}{Z}\right)\frac{f}{6H_0^2} + 2f',$$ (12)

where we are treating $f = f(6H_0^2 Z/(1-Z))$ as an arbitrary function of $Z$.

A dual interpretation of $f$ and $f'$ is always possible in the sense that they can be seen as functions of the scalar that governs the modified theory of gravity or as functions of $H$. Whatever the case, we will always present them as functions of $Z$. As a consequence the equation for $X_1 + X_2$ is also a function of $Z$ until any particular form of $f$ is specified. We can therefore use this equation to eliminate either $X_1$ or $X_2$, making the phase space two-dimensional.

Let us choose to eliminate $X_1$ and consider the evolution of $\{X_2, Z\}$,

$$\frac{dX_2}{dN} = \frac{3X_2}{m+1}\left(\frac{(w_2 - w_1)X_2}{f'} - (w_2 + 1)m - (w_1 + 1)n + 2w_1 - w_2 + 1\right),$$ (13)

$$\frac{dZ}{dN} = -\frac{3(Z-1)Z((w_1 + 1)(n - 2) + (w_1 - w_2)X_2/f')}{m+1},$$ (14)

where we have taken advantage of (12) and introduced the convenient functions

$$m(Z) := \frac{2Qf''}{f'} = \frac{12H_0^2 Zf''}{(1-Z)f'},$$ (15)

$$n(Z) := \frac{f}{Qf'} = \frac{f(1-Z)}{6H_0^2 Zf'}.$$ (16)

It is also worth keeping in mind that $f'$ is dimensionless whereas $f''$ has units of $H_0^{-2}$ because $f$ has dimensions $H_0^2$. For this reason, Equations (13) and (14) and $m(Z)$ and $n(Z)$ are dimensionless as well.

Once a specific theoretical setting is chosen through $f$ we are left with a closed system of equations ready to be studied. Fortunately, some of the key features of this set of equations do not depend on the chosen form of $f$, so a number of very broad conclusions may be drawn, which adds to the interest of our analysis and approach.

### 3.2. Fixed Points

For this discussion we assume $w_1 \neq w_2$ and $(w_1, w_2) \neq -1$, that is, two different fluids and neither of them is a cosmological constant. There are then two families of fixed points for the system, which we will look at individually.

The first one corresponds to the points $\{X_2, Z\} = \{0, Z^\star\}$, where the second coordinate is specified through solutions of the algebraic equation

$$\frac{(n(Z) - 2)(Z - 1)Z}{1 + m(Z)} = 0.$$ (17)

Note that the locations of these points are independent of both fluid parameters $w_1$ and $w_2$. The solutions where $n(Z^\star) = 2$ and $m(Z^\star) \to \infty$ with $n(Z^\star)$ finite will be particularly important. As such, we will name these points $P_n$ and $P_m$, respectively. In the following section we show that these points possess very particular fixed properties that are of interest when assessing the validity of different cosmological models.

The second family is characterised by the two points $B = \{X_2^\star, 1\}$ and $C = \{X_2^\star, 0\}$ with

$$X_2^\star = \frac{f'}{w_2 - w_1}\left((w_1 + 1)n(Z) + w_2 + (w_2 + 1)m(Z) - 1 - 2w_1\right),\tag{18}$$

where $X_2^*$ is evaluated at $Z = 1$ and $Z = 0$. Here the location does depend on the particular values of $w_1$ and $w_2$, and $X_2^*$ should be treated as a function of $Z$.

Note that the existence of the points and their belonging to the physical phase space is not guaranteed. To assess this one must study the Hubble constraint for that particular model, which can be written with the help of $n(Z)$ as

$$X_1 = f'(2 - n(Z)) - X_2.\tag{19}$$

We require that both variables $X_1$ and $X_2$ be positive in order to satisfy energy conditions, along with $0 \leq Z \leq 1$.

Lastly, we note the possibility that the dynamical equations diverge for some particular value of $Z$, which occurs if $m(Z) = -1$ or $f' = 0$. In these cases, trajectories cannot be extended beyond this $Z$ coordinate and the phase space exhibits a 'critical line' behavior, see [44]. For a full analysis including the existence criteria of the critical points, knowledge of the model $f(Q)$ is needed.

### 3.3. Physical Parameters of the General System

The deceleration parameter $q$ and the effective equation of state $w_{\text{eff}}$ can be expressed in terms of the dynamical variables

$$q := -\frac{\ddot{a}a}{\dot{a}^2} = -1 - \frac{3}{2}\frac{\left(X_1 + w_1 X_1 + X_2 + w_2 X_2\right)\left(n(Z) - 2\right)}{(X_1 + X_2)(m(Z) + 1)},\tag{20}$$

$$w_{\text{eff}} := \frac{p_{\text{tot}}}{\rho_{\text{tot}}} = -1 - \frac{\left(X_1 + w_1 X_1 + X_2 + w_2 X_2\right)\left(n(Z) - 2\right)}{(X_1 + X_2)(m(Z) + 1)},\tag{21}$$

where the total energy density $\rho_{\text{tot}}$ is defined as $\rho_{\text{tot}} = \rho_1 + \rho_2 + \rho_f$ with $\rho_f$ representing the additional non-GR terms

$$\rho_f := 3H^2 + \frac{1}{2}f - 6H^2 f'.\tag{22}$$

The total pressure is defined similarly $p_{\text{tot}} = p_1 + p_2 + p_f$. In Equations (20) and (21) the variable $X_1$ can equally be rewritten in terms of $X_2$ and $Z$ using the Friedmann constraint, but a full analysis cannot be carried out until a function $f$ is specified.

The density parameter of the additional non-GR terms will turn out to be useful later on, which we can express in terms of our dynamical variables as

$$\Omega_f := \frac{\rho_f}{3H^2} = 1 - f'(2 - n(Z)),\tag{23}$$

which satisfies $\Omega_1 + \Omega_2 + \Omega_f = 1$ from the Friedmann equation. Hence we have obtained a very neat expression representing the contributions of the modified theory beyond GR. As previously mentioned, a more standard dynamical systems formulation would introduce a dynamical variable for this $\Omega_f$ (e.g., the $Y$ in Section 2) but because it can be written totally in terms of $Z$ this is not necessary. We therefore obtain a phase space with fewer dimensions at the expense of a more cumbersome Hubble constraint.

For the fixed points $P_n$ satisfying $n(Z) = 2$ one immediately has $\Omega_f = 1$. Similarly for points $P_m$, looking at the definition of $m(Z)$ in (15), we see that if $m(Z) \to \infty$ whilst $n(Z)$ stays finite, we also obtain $\Omega_f = 1$. For these two solutions, the deceleration parameter and equation of state are fixed to be $q = w_{\text{eff}} = -1$, representing a de Sitter Universe. In fact, this is a necessary requirement for any de Sitter solution given that we have assumed $w_1 \neq w_2$ and $w_1, w_2 \neq -1$. Models where $n(Z) \neq 2$ and $m(Z) \nrightarrow \infty$ for some $Z$ in the range $(0, 1)$ cannot possess a de Sitter fixed point. This immediately rules out models such as $f(Q) \propto Q^\alpha$ for $\alpha \neq 1/2$, or $f(Q) = Q + \beta Q^2$ for $\beta \geq 0$.

For the other fixed points B and C at $\{X_2^\star, 1\}$ and $\{X_2^\star, 0\}$, we can also determine the deceleration parameter and effective equation of state. For both of these points one obtains $w_{\text{eff}} = w_2$ and $q = (1 + 3w_2)/2$, which is a remarkably general result independent from the model. These results are summarized in Table 1.

**Table 1.** Table of critical points with fixed values of deceleration parameter $q$ and effective equation of state $w_{\text{eff}}$.

| Point | $X_2$ | $Z$ | $q$ | $w_{\text{eff}}$ | Requirement |
|-------|-------|-----|-----|------------------|-------------|
| $P_m$ | 0 | $Z^\star$ | $-1$ | $-1$ | $n(Z^\star) = 2$ |
| $P_n$ | 0 | $Z^\star$ | $-1$ | $-1$ | $m(Z^\star) \to \infty$ |
| B | $X_2^\star$ | 1 | $\frac{1}{2}(1 + 3w_2)$ | $w_2$ | $X_2^\star$ evaluated at $Z = 1$ |
| C | $X_2^\star$ | 0 | $\frac{1}{2}(1 + 3w_2)$ | $w_2$ | $X_2^\star$ evaluated at $Z = 0$ |

Note that there may exist other fixed points at $\{0, Z^\star\}$, solutions to (17), which have not been included in the Table. This is because their properties are more dependent on the specific model and do not lead to fixed values of $q$ or $w_{\text{eff}}$. Furthermore, note that we have not yet fully discussed the existence conditions for the fixed points, and their presence in the physical phase space depends on the Hubble constraint for that particular model. We now move on to studying a chosen model where a full analysis can be carried out.

## 4. Applications to $f(Q)$ Models

### 4.1. Anagnostopoulos et al. Model

The power-exponential model proposed by Anagnostopoulos et al. in [51] displays a number of interesting features and was shown to pass a variety of observational tests [51,52]. In particular, the authors studied the model against Supernovae type Ia (SNIa), Baryonic Acoustic Oscillations (BAO), cosmic chronometers (CC), and Redshift Space Distortion (RSD) data and found that it is comparable, and for some datasets preferable, to the $\Lambda$CDM model. Moreover, it immediately passes early universe constraints. As such, it has been shown to be a genuine alternative to the $\Lambda$CDM concordance model and worthwhile studying from a dynamical systems perspective.

The model is given by the function

$$f(Q) = Q e^{\lambda \frac{Q_0}{Q}} , \tag{24}$$

with the single free parameter $\lambda$. A dynamical systems analysis was recently performed for this model in [53]. There the authors studied the background and perturbation equations of a universe with a single fluid matter component ($w = 0$), and the subsequent phase space was three-dimensional. It is interesting to then study this model in our reduced dimensionality formulation with an additional matter fluid component, which will turn out to be two-dimensional. Moreover, the reduced dimensions in our approach will turn out to make the stability analysis much simpler to compute.

In the limit that $\lambda$ vanishes the model (24) reduces to GR without a cosmological constant. It does not however have a direct $\Lambda$CDM limit. When the parameter $\lambda$ is small, to first order the function behaves like GR with a cosmological constant term $Q_0\lambda$, and this behavior will be observed in the phase space analysis. The sign of the parameter $\lambda$ leads to different phase spaces, and so we will investigate both cases. We will also assume that $\lambda \neq 0$, as this trivially leads back to GR.

The system is described by the dynamical Equations (13) and (14) along with Hubble constraint (19), with the functions $n(Z)$ and $m(Z)$ taking a remarkably simple form

$$n(Z) = \frac{Z}{Z + \lambda(Z - 1)} , \tag{25}$$

$$m(Z) = \frac{2(Z - 1)^2 \lambda^2}{Z(Z + \lambda(Z - 1))} . \tag{26}$$

The $f'$ term written explicitly in terms of the variable $Z$ is

$$f'(Q) = (Z + \lambda(Z-1)) \frac{e^{-\lambda(Z-1)/Z}}{Z} . \tag{27}$$

The fixed points are solutions to Equations (17) and (18), which can be easily solved using the exact forms of $n(Z)$, $m(Z)$ and $f'$ given above. The first family of solutions along the $X_2 = 0$ line with $Z = Z^*$ are the points A $= \{0,1\}$, $P_m = \{0,0\}$ and $P_n = \{0, 2\lambda/(1+2\lambda)\}$. Point A is an additional solution to the algebraic Equation (17) with properties that could not be determined in general, therefore it was left out of Table 1. The critical point $P_m$ satisfies $m(Z) \to \infty$ with $n(Z) = $ finite, whilst the point $P_n$ is a solution to $n(Z) = 2$. Hence these two points describe de Sitter attractors, as explained in the previous section and in Table 1.

The second set of solutions from Equation (18) include the point B $= \{1,1\}$ and a conditional point C at $\{0,0\}$ which requires $\lambda < 0$. However, we will ignore this final point because it coincides with $P_m$. Point B is the $\rho_2$ matter-dominated point. Note again that we have not yet assessed the validity of any of the fixed points; only those satisfying $(X_1, X_2) \geq 0$ and $1 \geq Z \geq 0$ are physically meaningful, for which we will need to use the Hubble constraint.

The Hubble constraint (19) can be written explicitly in terms of the variables as

$$X_1 + X_2 = e^{-\lambda(Z-1)/Z} \left(1 + \frac{2\lambda(Z-1)}{Z}\right). \tag{28}$$

The requirement that our matter fluids have positive energy density leads to physical bounds on the phase space. In particular, one notes that for positive $\lambda$ our fluid density parameters take the maximum value of one, whilst for negative $\lambda$ we instead obtain $(X_1, X_2) \leq \frac{2}{\sqrt{e}} \approx 1.21$. This situation, where the density parameters can be greater than one, can be understood in physical terms by considering the modified density parameter $\Omega_f$ in Equation (23). For positive $\lambda$ the density parameter is non-negative, and from the Hubble equation $\Omega_1 + \Omega_2 + \Omega_f = 1$ we can conclude that $\Omega_1 + \Omega_2 \leq 1$. However, for negative $\lambda$ we instead have the minimum of $\Omega_f = 1 - \frac{2}{\sqrt{e}}$, which leads to $\Omega_1 + \Omega_2 \leq \frac{2}{\sqrt{e}}$.

Using (28) we can determine $X_1$ at each of the fixed points, as well as the conditions for the point to be part of the physical phase space. The points A and B are always present irrespective of $\lambda$. The de Sitter points $P_m$ and $P_n$ require $\lambda < 0$ and $\lambda > 0$ respectively. It is also interesting to note that all of the fixed points, their locations and their existence criteria are independent of the specific fluid equation of state. The deceleration parameter and effective equation of state can be evaluated at each of the fixed points using Equations (20) and (21). Lastly, linear stability theory has been applied to the fixed points, which can be found in Appendix A. In this formulation points A, B and $P_n$ are hyperbolic and $P_m$ is nonhypebolic. However, we show in Appendix A that $P_m$ acts as the late-time attractor within the physical phase space ($\lambda < 0$). These results are collated in Table 2.

**Table 2.** Table of fixed points for the Anagnostopoulos et al. model.

| Point | $X_1$ | $X_2$ | $Z$ | $q$ | $w_{\text{eff}}$ | Existence Conditions | Stability |
|:---:|:---:|:---:|:---:|:---:|:---:|:---:|:---:|
| A | 1 | 0 | 1 | $\frac{1}{2}(1+3w_1)$ | $w_1$ | none | Saddle |
| B | 0 | 1 | 1 | $\frac{1}{2}(1+3w_2)$ | $w_2$ | none | Unstable |
| $P_m$ | 0 | 0 | 0 | $-1$ | $-1$ | $\lambda < 0$ | Nonhyperbolic |
| $P_n$ | 0 | 0 | $2\lambda/(1+2\lambda)$ | $-1$ | $-1$ | $\lambda > 0$ | Stable |

### 4.2. Phase Space Analysis

Next, we will fix the equations of state of our two fluid components to be $w_1 = 0$ and $w_2 = 1/3$, representing matter and radiation. The phase portraits for this model with positive and negative values of the free parameter $\lambda$ are shown in Figure 1. The absolute value of the free parameter is chosen to be $|\lambda| = 0.371$, as this was shown in [51] to be the

most consistent with observational constraints. The bordered region highlights the physical phase space and the red overlay represents regimes where the expansion of the Universe is accelerating $q < 0$.

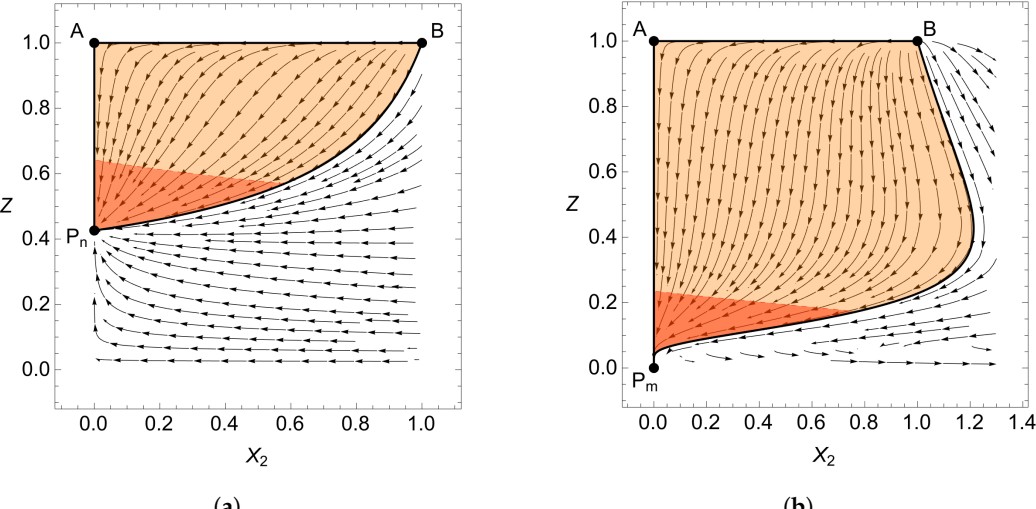

(a)

(b)

**Figure 1.** Phase portraits for the Anagnostopoulos et al. model (24). The physical phase space is within the bordered region whilst the red region represents accelerated expansion. (**a**) Phase space with positive $\lambda$. (**b**) Phase space with negative $\lambda$.

It is somewhat expected that for positive $\lambda$ the qualitative results should be the same as GR with a positive cosmological constant, as the dynamics are similar to $\Lambda$CDM for this parameter value [51,53]. Comparing the phase portrait in Figure 1a to that of GR with a cosmological constant, which can be found in our previous work using the same dynamical systems formulation [44], reveals that the qualitative features are indeed the same. The phase space contains an early radiation-dominated repeller, point B, a matter dominate saddle, point A, and a late-time de Sitter attractor, point $P_n$. Stability analysis indeed verifies that point B is unstable, A is a saddle and $P_n$ is stable.

For the case of a negative parameter $\lambda$, see Figure 1b, the phase space is similar but distinctly different. The fixed points of the system remain the same except point $P_m$ at $\{0, 0\}$ replaces $P_n$ at $\{0, 2\lambda/(1 + 2\lambda)\}$. The new de Sitter point $P_m$ possess the same properties as $P_n$, refer to Tables 1 and 2. The obvious difference is that the physical phase space extends outwards beyond $X_2 = 1$. As previously explained, this is due to the modified density parameter $\Omega_f$ having a negative lower bound for $\lambda < 0$. This leads to a noticeably different evolution of the density parameters and physical parameters $q$ and $w_{\text{eff}}$.

In Figure 2, the evolution of the matter and radiation density parameters $\Omega_m = X_1$, $\Omega_r = X_2$, the deceleration parameter $q$ and the effective equation of state $w_{\text{eff}}$ are shown for both phase spaces. Figure 2a shows the evolution for $\lambda > 0$ of a trajectory following a heteroclinic orbit from points B $\rightarrow$ A $\rightarrow$ $P_n$, whilst Figure 2b follows the heteroclinic orbit B $\rightarrow$ A $\rightarrow$ $P_m$ for $\lambda < 0$. The dashed lines represent the evolution of these parameters for GR with a positive cosmological constant, $f(Q) = Q + 2\Lambda Q_0 = 6H^2 + 12H_0^2\Lambda$ with $\Lambda = |\lambda|$. We have chosen $\Lambda$ to equal $\lambda$ such that the de Sitter point $P_n$ of GR and the de Sitter point $P_n$ of the Anagnostopoulos model (with positive $\lambda$) take the same values. It is interesting to note that for $f(Q) = Q + 2\Lambda Q_0$ the fixed points are exactly the A, B and $P_n$ given in Table 2 with $\lambda$ replaced by $\Lambda$. For the case where the free parameter $\lambda$ is negative, the same matching cannot be done because the point $P_m$ does not exist for GR with a cosmological constant. This can be seen from the function $n(Z) = 1 - 2\Lambda + \frac{2\Lambda}{Z}$ and $m(Z) = 0$. The point $P_m$ cannot exist because it requires $m(Z) \rightarrow \infty$, see [44] for more details.

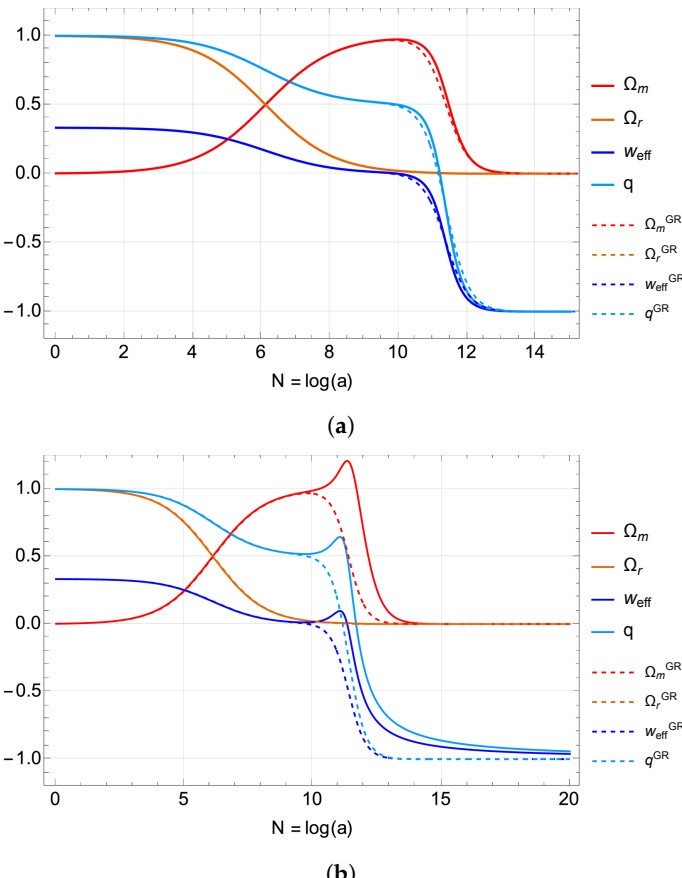

**Figure 2.** Evolution of density parameters $\Omega_m$, $\Omega_r$, effective equation of state $w_{\text{eff}}$, and deceleration parameter $q$ for GR with a positive cosmological constant (dashed) and Anagnostopoulos et al. model (solid line). (**a**) Evolution for positive $\lambda$. (**b**) Evolution for negative $\lambda$.

The background evolution of the model with positive $\lambda$ can be seen to match very closely with its GR counterpart, given the same initial conditions. For the negative $\lambda$ case in Figure 2b, a sharp spike in the matter density parameter can be noted before approaching the de Sitter point with $w_{\text{eff}} = -1$.

Overall, the dynamical system analysis gives a good understanding of the background dynamics of the Anagnostopoulos et al. model for both signs of the free parameter $\lambda$. In the positive $\lambda$ case, the fixed points and stability of the system match what is found for GR with a positive cosmological constant. In fact, from a qualitative point of view, these models are identical. For the negative $\lambda$ case, the fixed points of the system have the same properties but it is interesting to note the different physical phase space as well as the different evolutions for the density parameters.

## 5. Summary

We have presented a dynamical systems formulation that is well suited to the cosmological equations arising in various modified gravity theories, with $f(Q)$ being the focus of this work. Once a model function $f$ has been specified, the drawback of a more complicated constraint equation is indeed present but of little significance. This is especially true due to the clarity gained when dealing with a two-dimensional as opposed to a three-dimensional phase space (and similarly for higher-dimensional analogs). This same approach can easily be generalized to include additional degrees of freedom represented by additional dynamical variables. For example, the inclusion of extra matter sources, scalar fields, or non-zero spatial curvature could be easily realized within our formulation, see for instance [38]. The use of the functions $m(Z)$ and $n(Z)$ introduced in Equation (15) and (16) is particularly adept at assessing the validity of models, as the existence of late-time de Sitter points can be established almost immediately.

For the recently proposed $f(Q)$ model by Anagnostopoulos et al. [51], we analyzed the phase space for a universe comprised of two fluid components, matter and radiation. This complements the previous dynamical systems analysis performed on this model for matter and matter perturbations for a positive value of the free parameter [53]. Indeed, for positive $\lambda$ we reproduced the dynamics of $\Lambda$CDM. This is perhaps to be expected from the series expansion of the function $f$ for small $\lambda$, with the leading order terms being $Q + Q_0\lambda$, which is exactly the Lagrangian of GR plus a cosmological constant.

However, we also find the surprising result that a negative value of the parameter leads to a qualitatively similar dynamical system. In particular, it is interesting to note that a negative value of the parameter $\lambda$ in fact still acts as a positive cosmological constant, leading to a late-time de Sitter point within the phase space. This was determined by studying the functions $n(Z)$ and $m(Z)$, Equation (25) and (26), which took a remarkably simple form for this model.

The results of the dynamical systems analysis for the Anagnostopoulos et al. [51] model show that at the background level, it passes cosmological observational constraints, displaying the correct evolutionary behavior of the matter density parameters and the effective equation of state. Namely, for any non-zero value of the parameter $\lambda$, there exists an early-time radiation-dominated point, a matter saddle, and an accelerating de Sitter attractor. This study, the first to use both matter and radiation sources, gives more reason to continue to investigate this model in the future.

In summary, the approach taken leads to a number of model-independent results, which would be especially interesting to investigate in more detail. The moral behind the approach can ultimately be traced to the dynamical systems formulations of GR: for each matter source $\rho_i$ we introduce the corresponding density parameter $\Omega_i$ as a variable. We then introduce one additional variable related to the remaining terms in the Hubble constraint in order to close the system, for which we chose the Hubble function $H$. In second order modifications, such as $f(T)$ and $f(Q)$ gravity, we have shown that this same prescription works for all models. This is in contrast to most of the dynamical systems formulations used in modified gravity. In the future it would be interesting to further study promising alternatives to the $\Lambda$CDM model using such a formulation. It would also be interesting to search for models that satisfy current observations yet exhibit a different and more complex fixed point behavior to GR, which could lead to qualitatively different predictions.

**Author Contributions:** Conceptualization, C.B. and R.L.; methodology, validation, formal analysis and investigation, writing—original draft preparation, writing—review and editing, C.B., R.L. and E.J.; visualization, C.B. and E.J.; funding acquisition, R.L. and E.J. All authors have read and agreed to the published version of the manuscript.

**Funding:** R.L. was supported by the Spanish Ministry of Science and Innovation through research projects PID2021-123226NB-I00 (comprising FEDER funds), and also by the Basque Government and Generalitat Valenciana through research projects IT1628-22 and PROMETEO/2020/079, respectively. Erik Jensko is supported by EPSRC Doctoral Training Programme (EP/R513143/1).

**Data Availability Statement:** No data were created.

**Acknowledgments:** R.L. thanks José Beltrán Jiménez for conversations.

**Conflicts of Interest:** The authors declare no conflict of interest. The funders had no role in the design of the study; in the collection, analyses, or interpretation of data; in the writing of the manuscript; or in the decision to publish the results.

## Appendix A. Stability Analysis of Anagnostopoulos et al. Model

Here we apply linear stability theory to each of the fixed points in Table 2 for the model $f(Q) = Q \exp(\lambda Q/Q_0)$ considered in Section 4. Where linear stability theory fails, we look to see what can be said about the nature of the fixed points by examining the autonomous equations and constraints directly.

For point A $\{0,1\}$ we obtain the eigenvalues

$$\lambda_1^A = 3(w_1 - w_2) \quad , \quad \lambda_2^A = 3(1 + w_1). \tag{A1}$$

The point is therefore a saddle for $w_1 < w_2$ or unstable for $w_1 > w_2$, as the second eigenvalue is always positive due to the assumption that $w_1 > -1$. Point B $\{1,1\}$ has eigenvalues

$$\lambda_1^{\mathrm{B}} = 3(w_2 - w_1) = -\lambda_1^{\mathrm{A}} \quad , \quad \lambda_2^{\mathrm{B}} = 3(1 + w_2) , \tag{A2}$$

which is unstable for $w_1 < w_2$ or a saddle point for $w_1 > w_2$. Again, the second eigenvalue is always positive. Due to the freedom in the ordering of our matter fluids $\rho_1$ and $\rho_2$, we choose $w_1 < w_2$ without loss of generality such that point A is a saddle and point B is unstable.

Point P$_n$ $\{0, 2\lambda/(1 + 2\lambda)\}$ has eigenvalues

$$\lambda_1^{\mathrm{P_n}} = -3(1 + w_1) = -\lambda_2^{\mathrm{A}} \quad , \quad \lambda_2^{\mathrm{P_n}} = -3(1 + w_2) = -\lambda_2^{\mathrm{B}} , \tag{A3}$$

and is therefore always stable. This is the stable late-time de Sitter point of the system.

Point P$_m$ has eigenvalues

$$\lambda_1^{\mathrm{P_m}} = 0 \quad , \quad \lambda_2^{\mathrm{P_m}} = -3(1 + w_2) = -\lambda_2^{\mathrm{B}} , \tag{A4}$$

therefore methods beyond linear stability theory must be used to fully determine the stability.

A closer look at the nonhyperbolic point is shown in Figure A1, as well as the physically allowed values of $X_2$ and $Z$ as determined from the Hubble constraint (28). Recall that we require $\lambda < 0$ for the existence of this point. Despite the fact that the point appears to be mathematically unstable, with trajectories moving away from the point in Figure A1, the Hubble constraint can be used to determine the fate of trajectories within the physical phase space. The boundary of the physical phase space is described by the equations

$$X_2 = e^{\lambda\left(\frac{1}{Z} - 1\right)} \left(1 + \frac{2\lambda(Z - 1)}{Z}\right) \quad , \quad \text{with } X_2 \geq 0 \, , \, 0 \leq Z \leq 1 \, , \, \lambda < 0 . \tag{A5}$$

As $Z$ approaches zero (from above) $X_2$ goes to zero. One can also show that for all $X_2 \geq 0$ trajectories always travel in the negative $Z$ direction whilst $Z$ is between 0 and 1. This can be most easily seen by substituting the expression for $X_2$ on the physical boundary (A5) into the autonomous equation $dZ/dN$. This resulting equation is

$$\frac{dZ}{dN} = -\frac{3(1 + w_2)Z^2(1 - Z)\big(Z - 2\lambda(1 - Z)\big)}{Z^2 + 2\lambda^2(Z - 1)^2 - \lambda Z(1 - Z)} , \tag{A6}$$

where the equation of state $w_2 > -1$. All terms in the numerator and denominator are positive for $\lambda < 0$, therefore $dZ/dN$ is negative and trajectories on the boundary approach the origin.

Following the same logic, the same result can be shown for the general equation $dZ/dN$ with $\lambda < 0$. We can therefore conclude that all physical trajectories satisfying the Hubble constraint travel towards and terminate at the origin, point P$_m$. This is because trajectories do not cross the boundary, and must end at $Z = 0$ which is only allowed at $X_2 = 0$. This indeed matches what can be seen from the phase portraits, Figures 1b and A1, and the numerical solutions in Figure 2b.

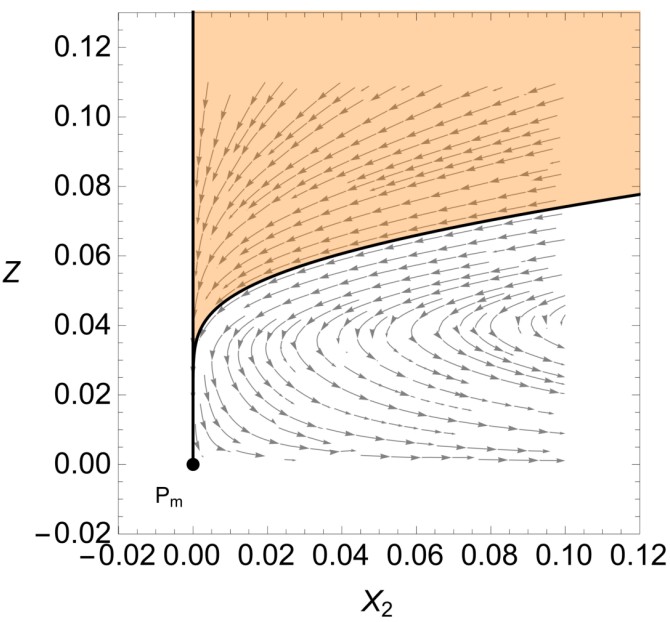

**Figure A1.** Nonhyperbolic fixed point P$_m$.

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
