# Peer review of "Dynamical Systems Analysis of f(Q) Gravity"

_universe, doi:10.3390/universe9040166_

Round 1

Reviewer 1 Report

Reviewer's comments

Manuscript details:
Journal: Universe
Manuscript ID: universe-2228184
Type of manuscript: Article
Title: Dynamical systems analysis of $f(Q)$ gravity
Authors: Christian Boehm
er, Erik Jensko, Ruth Lazkoz
Special Issue: Modified Gravity Approaches to the Tensions of ΛCDM
https://www.mdpi.com/journal/universe/special_issues/Gravit

Modified gravity theories can be used for the description of homogeneous and isotropic cosmological models using the corresponding field equations. In these cases the field equations become autonomous differential equations because in such models all variables only depend on cosmological time, or some another suitably chosen time parameter and the field equations can always be cast into the form of a dynamical system to study such models.

Authors study the f(Q) modified gravity models through a model independent set of variables. They claim that through a model of independent set of variables they are able to draw a scheme that is valid all f(Q) modified gravity models. Also, authors discuss the power-exponential model by Anagnostopoulos and perform phase space analysis for a universe comprised of two fluid components, matter and radiation. For positive lambda authors reproduced the dynamics of ΛCDM. They approach leads to a number of model-independent results. Also, the drawback of their approach is an additional level of complexity in the constraint equation, but on the other hand they formulated the dynamical system in fewer dimensions than usual.

My opinion is that this paper is interesting and useful for the potential readers and I would like to recommend publication of this paper in journal Universe.

Minor technical errors:

1. Problem with order of calling references in paper should be repaired. For example after references [22–26, 29–36, 40–42] for the first time appear references [27, 28, 38], ...

Author Response

As requested by the referee some citation numbering issues have been fixed

Reviewer 2 Report

In the manuscript the authors investigate, by using methods from the qualitative analysis of dynamical systems, a particular model of the f(Q) gravity theory. A number of model independent results are obtained, and the relation  of the model  with the \Lambda CDM paradigm is also considered. The manuscript may be publishable in Universe if the authors would fully consider the following points:

1. I think that the structure of the Introduction Section is not particularly satisfactory, and the A, B..., subsections look more like distinct topics linked artificially together. I recommend to the authors to reorganize/rewrite this section, by clearly stating, among others, the main goals of this study, and the achieved results.

2. The authors consider the specific model (29), and claim that it "....is of particular interest." Could they elaborate more on the importance of this model?

3.  The Summary Section of the manuscript should be extended, with more relevant discussions on the significance of the obtained results added.  

Author Response

Following this reviewers comments  the Introduction section  has been rewritten to make it more concise, homegeneous and straight to the point.  In the vein ogf this referee's commente we have also added more motivation for the particular f(Q) model studied and have made our conclusions longer and denser.

Reviewer 3 Report

Review Report

The authors have considered f(Q) gravity model, in particular the power-exponential model and generalized the type of fluids contained in these universes to arbitrary equations of state. The work carried out is very interesting. For the better understanding of the readers, the authors should include some past works on FRW model in other modified gravity theories in detail, in the Introduction section. For example: International Journal of Geometric Methods in Modern Physics Vol. 16, No. 2 (2019) 1950024; International Journal of Geometric Methods in Modern Physics; Vol. 18, No. 9 (2021) 2150134; Physics of the Dark Universe 30 (2020) 100618; Chinese Journal of Physics 66 (2020) 787–799. After the improvements, the paper can be reconsidered.

Author Response

We appreciate the referee's effort to point to us a collection of papers of specific modified gravity topics which may be tangentially related to our work as they will definitely broaden our view of the field as a whole. However, including these references would demand in fairness the addition of many other works by other authors working in the same side of the spectrum. Unfortunately we feel these would make the reference list too broad and less focused than it goes with the spirit of the invitation we received from the main editors. We hope the referee can appreciate this point and we thank him/her for helping us gather more knowledge about this common ground of interest.

Round 2

Reviewer 2 Report

The authors have improved their manuscript, and hence I think that the present version is suitable for publication in Universe. 

Reviewer 3 Report

The paper is not improved along the points suggested in the previous report.  If the authors include those points, then it can be recommended for publication.